# Sniper GMMs: Structured Gaussian mixtures poison ML on large $n$ small $p$ data with high efficacy

## Abstract

We propose a method for structured learning of Gaussian mixtures with low KL-divergence from target mixture models that in turn model the raw data. We show that samples from these structured distributions are highly effective and evasive in poisoning training datasets of popular machine learning training pipelines such as neural networks, XGBoost and random forests. Such attacks are especially destructive given the current uptrends towards distributed machine learning with several untrusted client devices that provide their data to servers and cloud service providers for privacy preserving distributed machine learning. In current day and age of machine learning, Gaussian mixtures are perceived to be an older/classical technique in practice, although they are still actively studied from a theoretical perspective. Therefore it is quite interesting to see that they can be highly effective in performing data poisoning attacks on complex ML pipelines if learned with the right structural constraints.

## 1   Introduction

Data poisoning attack methods have propped up in plenty [1, 2] to damage the efficacy of training machine learning models. Their mode of operation is based on either modifying existing training data records via attacks such as one pixel attacks [3] or via addition of a subsample of poisoned data points [4] to the training datasets. These methods attempt to evade detection by models that screen the datasets or ML pipelines and anomaly detectors to detect data poisoning. Post the filtering of any detected points (typically with false alarms or false negatives); the rest of the undetected points produce a degradation in model performance on otherwise genuine data points upon which model predictions are to be obtained post deployment of the model. These methods are currently based on adversarial training [5, 6, 7, 8, 9, 10, 11, 12, 13]. We provide an alternative attack scheme for data poisoning that is instead based on structured learning of Gaussian mixtures with low KL-divergence from target mixture models that in turn model the raw data. We showthat samples from these structured distributions are highly effective and evasive inpoisoning training datasets of popular machine learning training pipelines suchas neural networks, XGBoost and random forests. In current day and age of machine learning Gaussian mixtures are perceived to be an older/classical technique. Therefore it is quite interesting to see that they can be highly effective in performing data poisoning attacks if learned with the right structural constraints.

## 2   Structured Decoy Distribution Learning

We now present our proposed results that help in structured distribution learning of Gaussian mixtures such that the KL-divergence between the learnt mixture and the target mixture is minimized. This helps in learning distributions from which the poisoned data points can be sampled from. The motivation is to use RKHS and distance based statistical dependency measures such as distance correlation, HSIC, MMD [14] between multivariate Gaussians as a gadget to minimize KL-divergence between Gaussian mixtures. Therefore we first start by connecting distance correlation to KL-divergence in the case of multivariate Gaussians as follows.

**Theorem 2.1** (Separability theorem). *Minimization of distance correlation* $\operatorname*{argmin}_{\mathbf{Z}}(\mathbf{X}, \mathbf{Z})$ *with respect to* $\mathbf{Z}$ *maximizes the Kullback-Leibler divergence,* $KL(\mathbf{X}||\mathbf{Z})$ *for* $\mathbf{X} \sim \mathcal{N}(0, \mathbf{\Sigma_X})$ *and* $\mathbf{Z} \sim \mathcal{N}(0, \mathbf{\Sigma_Z})$

32nd Conference on Neural Information Processing Systems (NIPS 2018), Montréal, Canada.

*Proof.* Distance correlation can be represented as $\frac{\text{Tr}\,(\mathbf{X^T X Z^T Z})}{\sqrt{\text{Tr}\,(\mathbf{X^T X})^2\,\text{Tr}\,(\mathbf{Z^T Z})^2}}$ [15]. For covariance matrices $\Sigma_{\mathbf{X}} = \mathbf{X^T X}$ and $\Sigma_{\mathbf{Z}} = \mathbf{Z^T Z}$ we have

$$\det[(\mathbf{X^T X})^2]\det([\mathbf{Z^T Z}])^2 \le \text{Tr}\,(\mathbf{X^T X Z^T Z}) \tag{1}$$
$$\le \sqrt{\text{Tr}\,(\mathbf{X^T X})^2\,\text{Tr}\,(\mathbf{Z^T Z})^2}$$

by arithmetic-geometric mean inequality for the lower bound and Cauchy-Schwartz inequality for the upper bound on distance covariance $\text{Tr}\,(\mathbf{X^T X Z^T Z})$. $\log\det(\mathbf{Z^T Z})$ is the differential entropy $h(\mathbf{Z})$ upto a constant for multivariate Gaussians. Similarly, the joint entropy $h(\mathbf{X}, \mathbf{Z})$ is given by $\log\det(\mathbf{\Sigma})$ where $\mathbf{\Sigma} = \left[\begin{array}{c|c} \mathbf{X^T X} & \mathbf{X^T Z} \\ \hline \mathbf{Z^T X} & \mathbf{Z^T Z} \end{array}\right]$ Kullback-Leibler divergence is defined using joint entropy and entropy as $h(\mathbf{X}||\mathbf{Z}) = h(\mathbf{X}, \mathbf{Z}) - h(\mathbf{X})$. By Fischer's inequality, we have

$$\det(\mathbf{\Sigma}) \le \det(\mathbf{X^T X})\det(\mathbf{Z^T Z})$$

As $\det(\mathbf{X^T X})$ is fixed and $\det(\mathbf{Z^T Z})$ decreases with decrease in distance covariance, an increase of $h(\mathbf{X}||\mathbf{Z})$ is only possible when $h(\mathbf{X}, \mathbf{Z}) = \log\det(\mathbf{\Sigma})$ increases which is inturn only possible when $\text{Tr}(\mathbf{X^T Z})$ decreases. Thereby minimizing sum of distance covariance and $\text{Tr}(\mathbf{X^T Z})$ maximizes the Kullback-Leibler divergence in the direction stated above while it also minimizes differential entropy $\det(\mathbf{Z^T Z})$. $\qquad\square$

**Distance correlation-KL divergence separability theorem:** We now plan to exploit our separability theorem we presented above given the fact that KL-divergence between Gaussian mixtures is separable into terms that only depend on the KL-divergence between the multivariate Gaussian components that form the mixture. We can thereby substitute $-D_{KL}(f_a||f_\alpha)$ with distance correlation $\text{DCor}(\Sigma_a^f, \Sigma_\alpha^f)$ and $-D_{KL}(f_a||g_b)$ with $\text{DCor}(\Sigma_a^f, \Sigma_b^g)$ instead based on this theorem which shows that optimizing KL-divergence between multivariate Gaussians is equivalent to optimizing distance correlation for the same.

## 2.1 Bounds on KL-Divergence between two Gaussian Mixtures

For the distribution learning problem motivated in the previous section, the key is to be able to learn a $\tau$-close Gaussian mixture to a given target Gaussian mixture. We therefore share some results on KL-divergences between Gaussian mixtures [16]. This helps exploit lower bounds in distribution testing problems that attempt to distinguish two distributions based on their samples. Let $f$ and $g$ be two PDFs in $\mathbb{R}^d$, where $d$ is the dimension of the observed vectors x. The KL-divergence between $f$ and $g$ is defined as $D_{KL}(f||g) = \int_{\mathbb{R}^d} f(x)\log\frac{f(x)}{g(x)}dx$. When $f$ and $g$ are PDFS of multivariate normals:

$$D_{KL}(f||g) = \frac{1}{2}\log\frac{|\Sigma^g|}{|\Sigma^f|} + \frac{1}{2}\text{Tr}((\Sigma^g)^{-1}\Sigma^f) + \frac{1}{2}(\mu^f - \mu^g)^T(\Sigma^g)^{-1}(\mu^f - \mu^g) - \frac{d}{2} \tag{2}$$

When $f$ and $g$ are PDFs for GMMs, the expression for $f$ is (with an analogous expression for $g$):

$$f(x) = \sum_{a=1}^{A}\omega_a^f f_a(x) = \sum_{a=1}^{A}\omega_a^f N\left(x; \mu_a^f, \Sigma_a^f\right) \tag{3}$$

A practical upper-bound on KL-divergence between two Gaussian mixtures is given by

$$D_{\text{avg}}(f||g) = \frac{1}{2}\sum_a \omega_a^f\left[\log\sum_\alpha \omega_\alpha^f e^{-D_{KL}(f_a||f_\alpha)} + \log\sum_\alpha \omega_\alpha^f z_{a\alpha} - \log\sum_b \omega_b^g t_{ab} - \log\sum_b \omega_b^g e^{-D_{KL}(f_a||g_b)}\right]$$

as detailed in the Appendix B.

# 3 Modified EM algorithm for our structured distribution learning problem

Therefore upon applying our separability theorem we have the following objective that needs to be minimized instead, as long as the initialization for the optimization is done such that the absolute value of the sum of two of the four terms above that do not depend on the target distribution are much higher than the absolute value of the sum of the other two terms which are known before hand. Upon invoking the separability theorem in 2.1, in order to minimize the above average bound on KL-divergence $D_{\mathrm{avg}}(f||g)$ between Gaussian mixtures, the following has to be minimized

$$\frac{1}{2}\sum_a \omega_a^f \left[ \log \sum_\alpha \omega_\alpha^f e^{\mathrm{DCov}(\Sigma_a^f,\Sigma_\alpha^f)} + \log \sum_\alpha \frac{\omega_\alpha^f}{\sqrt{|\Sigma_a^f + \Sigma_\alpha^f|}} - \log \sum_b \omega_b^g e^{\mathrm{DCov}(\Sigma_a^f,\Sigma_b^g)} - \log \sum_b \frac{\omega_b^g}{\sqrt{|\Sigma_a^f + \Sigma_b^g|}} \right] \tag{4}$$

This is fortunately possible because the KL divergence between Gaussian mixtures is expressed via separable terms of KL between components of Gaussian mixtures. Note that two terms are constant in here with respect to the target mixture distribution as follows

$$D_{\mathrm{avg}}(f||g) = \frac{1}{2}\sum_a \omega_a^f \left[ C_1 + C_2 - \log \sum_b \frac{\omega_b^g}{\sqrt{|\Sigma_a^f + \Sigma_b^g|}} - \log \sum_b \omega_b^g e^{\mathrm{DCov}(\Sigma_f^a,\Sigma_b^g)} \right] \tag{5}$$

With $\Sigma_g^f = \frac{1}{N-1}Z_b^T Z_b$, where $N$ is the number of samples, our problem is equivalent to minimizing the following for each component $a$

$$\omega_a^f \log \sum_\alpha \omega_\alpha^f e^{\mathrm{DCov}(\Sigma_a^f,\Sigma_\alpha^f)} + \omega_a^f \log \sum_\alpha \frac{\omega_\alpha^f}{\sqrt{|\Sigma_a^f + \Sigma_\alpha^f|}} \tag{6}$$

$$- \omega_a^f \log \sum_b \omega_b^g e^{\mathrm{DCov}(\Sigma_a^f,\frac{1}{N-1}Z_b^T Z_b)} - \omega_a^f \log \sum_b \frac{\omega_b^g}{\sqrt{|\Sigma_a^f + \frac{1}{N-1}Z_b^T Z_b|}} \tag{7}$$

$$= \omega_a^f (C_1 + C_2) - \omega_a^f \log \sum_b \omega_b^g e^{\mathrm{DCov}(\Sigma_a^f,\frac{1}{N-1}Z_b^T Z_b)} - \omega_a^f \log \sum_b \frac{\omega_b^g}{\sqrt{|\Sigma_a^f + \frac{1}{N-1}Z_b^T Z_b|}} \tag{8}$$

$$- \omega_a^f \log \sum_b \omega_b^g e^{\mathrm{DCov}\left(\Sigma_a^f,\frac{1}{N-1}(Z_b-\mu_b^g)^T(Z_b-\mu_b^g)\right)} - \omega_a^f \log \sum_b \frac{\omega_b^g e^{-\frac{1}{2}\left(\mu_b^g-\mu_a^f\right)^T\left(\Sigma_a^f+\frac{1}{N-1}(Z_b-\mu_b^g)^T(Z_b-\mu_b^g)\right)^{-1}\left(\mu_b^g-\mu_a^f\right)}}{\sqrt{|\Sigma_a^f + \frac{1}{N-1}(Z_b-\mu_b^g)^T(Z_b-\mu_b^g)|}} \tag{9}$$

$$+ \omega_a^f (C_1 + C_2) + \lambda.\mathrm{EMLoss}$$

where the EMLoss in the last term is the standard EM loss. Here, the objective function is regularized with the standard loss used in EM-algorithms for estimating Gaussian mixtures. Therefore we now have a modified EM algorithm that learns Gaussian mixtures with respect to a target distribution while satisfying the closeness constraints with respect to KL-divergence.

# 4 Modified EM algorithm for structured learning of Gaussian mixtures

**E-step updates**: For each component $b$ at step $t$, compute

$$\gamma_{ib}^{(t+1)} = \frac{\omega_b^{g\,(t)} p\left(y_i|\mu_b^{g\,(t)},\Sigma_b^{g\,(t)}\right)}{\sum_{b'=1}^B w_{b'}^{g\,(t)} p\left(y_i|\mu_{b'}^{g\,(t)},\Sigma_{b'}^{g\,(t)}\right)}, \quad i = 1,\dots,N$$

and finally

$$n_b^{(t+1)} = \sum_{i=1}^N \gamma_{ib}^{(t+1)}$$

| Dataset | Sample Size | Attributes | Balanced | # of Classes |
|---|---|---|---|---|
| EEG Eye State | 14,980 | 15 | Yes | 2 |
| Avila | 20,867 | 10 | Yes | 12 |
| Skin Segmentation | 245,057 | 4 | No | 2 |

Table 1: A listing of datasets that we used for empirical investigations is provided in this table along with their dimensions.

**M-step updates**: For each component $b$, compute the following update

$$\omega_b^{g(t+1)} = \frac{n_b}{N}$$

The rest of updates for the mean vector and covariances are in Appendix B.

**Theorem 4.1.** *The function* $\log \sum_b \frac{\omega_b^g}{\sqrt{|\Sigma_a^f + \Sigma_b^g|}}$ *is convex if*

$$\omega_b^g \sum_b \left( \frac{\omega_b^g}{\sqrt{|\Sigma_a^f + \Sigma_b^g|}} - \omega_b^g \right) \geq 0$$

*as this results in a positive semi-definite Hessian.*

*Proof.* The proof is in Appendix C. □

**Theorem 4.2.** *The function* $LogSumExp(p) = log(\sum_i(e_i^p))$ *is convex.*

*Proof.* The proof is in Appendix D. □

## 5 Upper and lower bounds on distance correlation

In the spirit of upper and lower bounds of [16] on KL-divergence that proved quite useful in this work, we propose our derived upper and lower bounds on distance correlation that we present in the Appendices F & G below.

## 6 Numerical Experiments

We performed numerical experiments on 3 UCI-ML repository datasets of EEG eye state, occupancy and Avila with their dimensions and specifications detailed in Table 1 above. We show in a series of captioned figures in the appendix below that the well-tuned classification models such as neural networks with increasing hidden layers of 1, 4 , 8 and 12 as well as models such as XGBoost and Random Forests cannot distinguish between the real and poisoned samples generated by our scheme, thereby making it really hard for an attacker that is dependent on machine learning to estimate the pair of mixture distributions used to model the real samples and to obtain poisoned samples respectively. Our pipeline consists of a model to detect a decoy Vs. non-decoy and in addition we also perform a label reconstruction attack to reconstruct the raw labels of the client. The poisoned samples are generated only using raw features. We see a spin-off empirical benefit that upon adding poisoned samples, not only do we prevent their detection; but we also make it extremely hard for the attacker to be able to reconstruct the raw labels corresponding to the raw data; via a second model. We use default SciPy parameters for powell minimization to optimize $mu$ and parameters of `ftol = 0.001,` `xtol = 0.001, maxfev = 4000` for optimizing $\mathbf{Z_b}$ in our modified EM algorithm while the rest of the steps in our algorithm are trivial to compute.

## 7 Conclusion

We show the efficacy and evasiveness of data poisoning with structured learning of Gaussian mixtures with low KL-divergence from target mixture models that in turn model the raw data. We also provide new results connecting RKHS and distance statistics like distance correlation to information theoretic measures like KL-divergence, and employ these results in optimizing for KL-divergence between Gaussian mixtures.

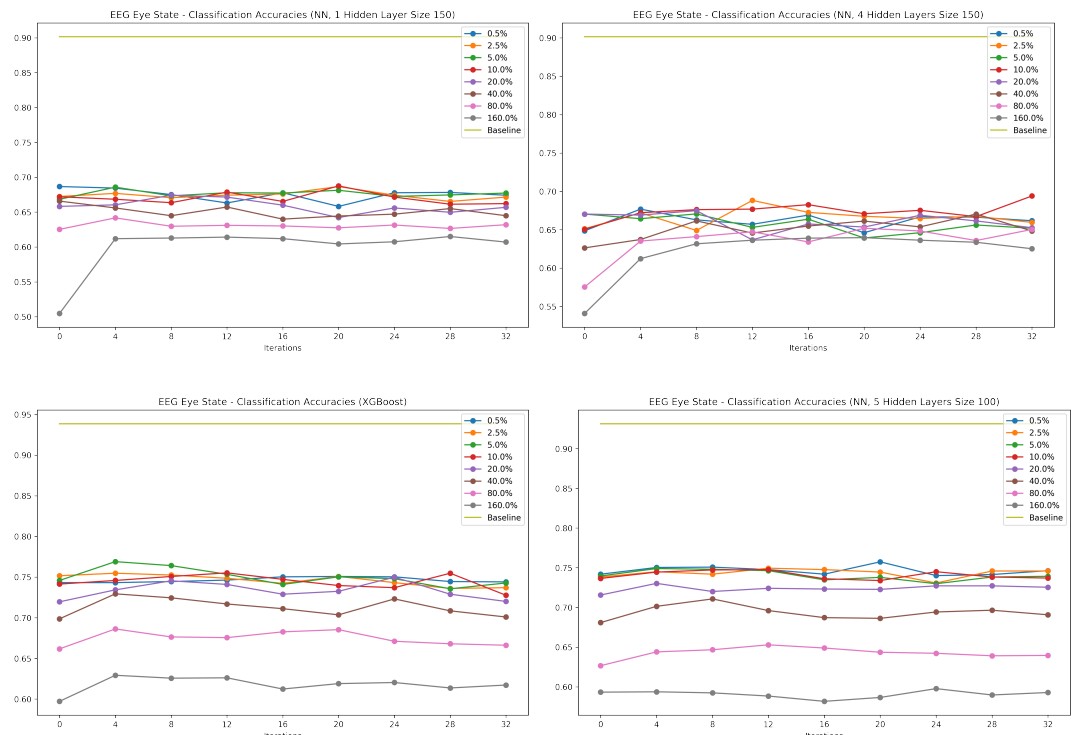

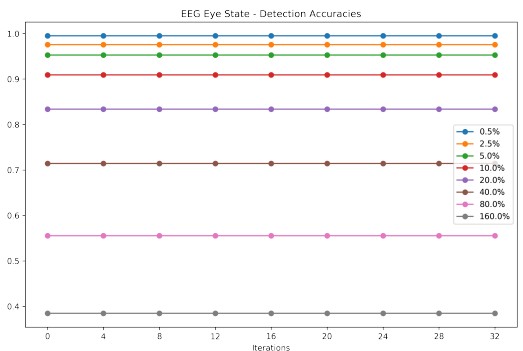

Figure 1: EEG: Classification of decoy Vs. non-decoy splinters using NN's, XGBoost and Random Forest shows that the models are unable to distinguish them when the sample size of decoy splinters is twice that of the non-decoy splinters. Our pipeline is a standard one used in data-poisoning schemes with two models; one to detect and one to classify. We obtain similar results upon using anomaly detectors such as isolation forests as well. The pipeline consists of a model to detect a decoy Vs. non-decoy and in addition we also perform a label reconstruction attack to reconstruct the raw labels of the client. The splinters are generated only using raw features. We see a spin-off empirical benefit that upon adding decoy splinters, not only do we prevent their detection; but we also make it extremely hard for the attacker to be able to reconstruct the raw labels corresponding to the raw data; via a second model.

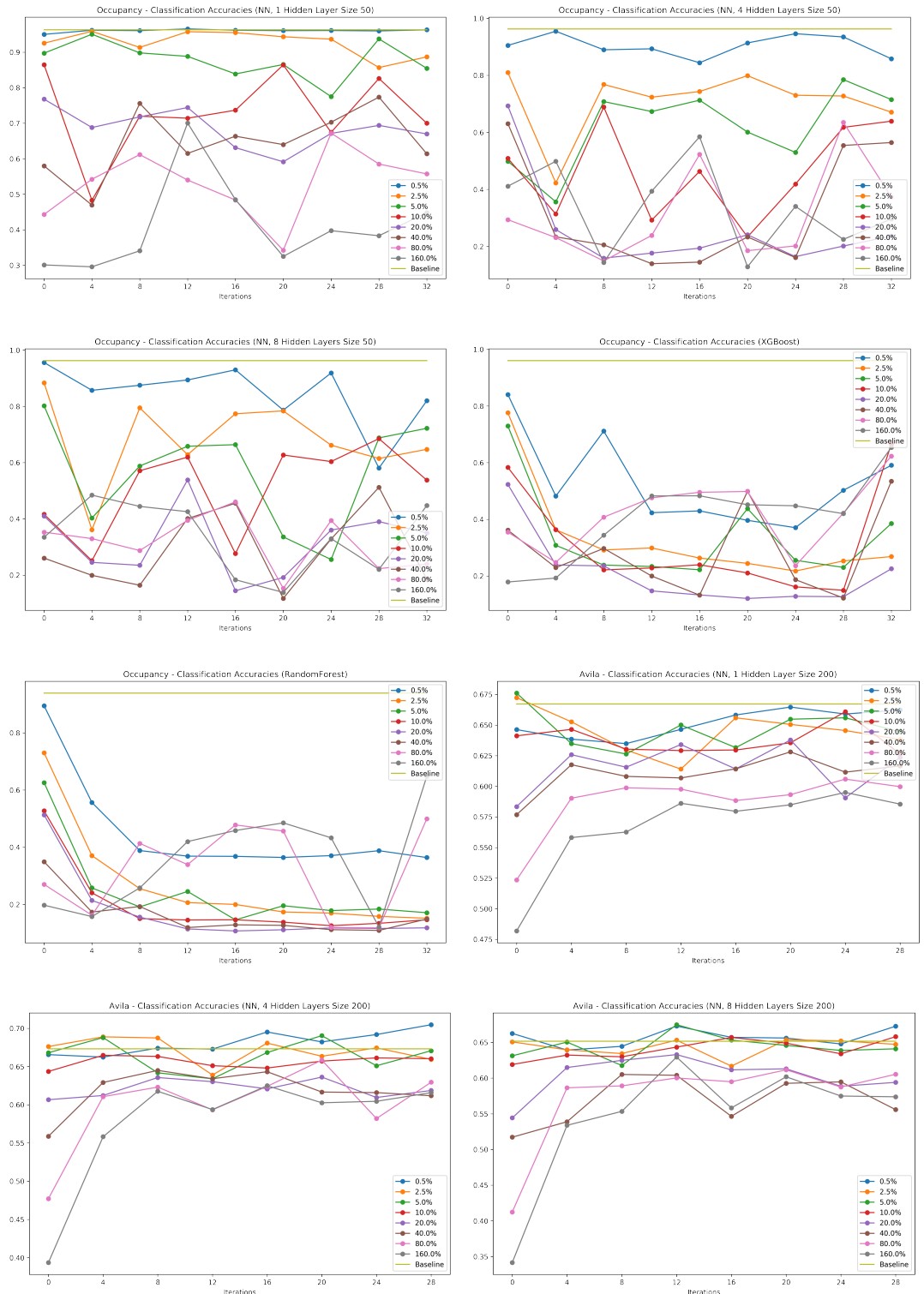

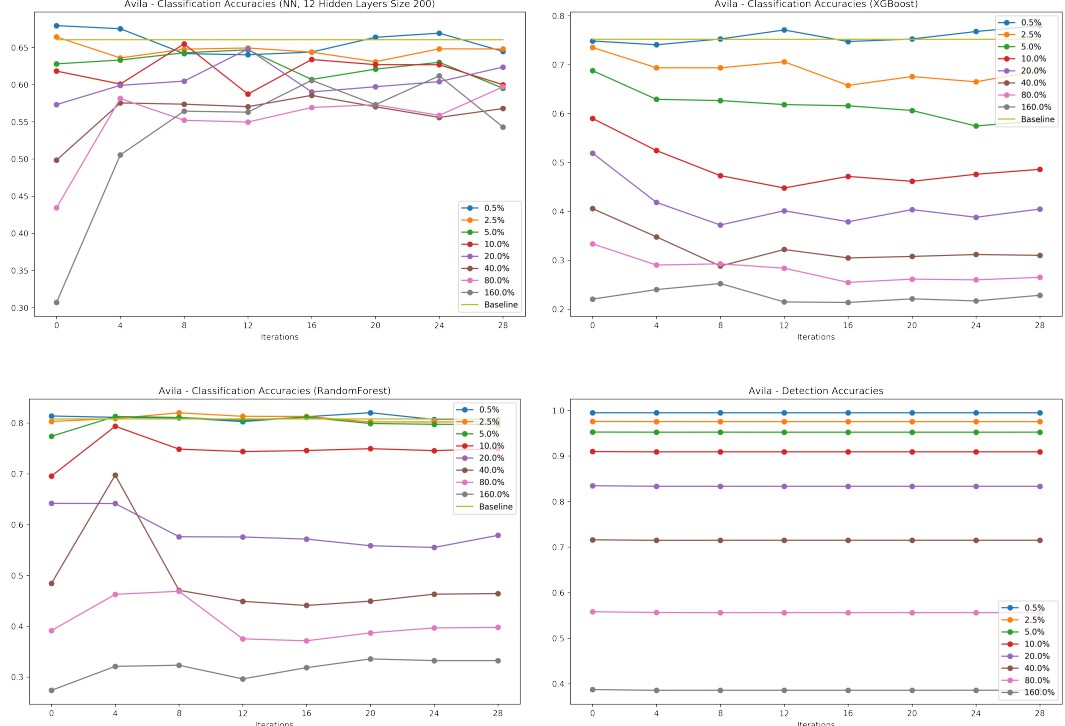

Occupation and Avila datasets: Classification of decoy Vs. non-decoy splinters using NN's, XGBoost and Random Forest shows that the models are unable to distinguish them when the sample size of decoy splinters is twice that of the non-decoy splinters. Our pipeline is a standard one used in data-poisoning schemes with two models; one to detect and one to classify. Our pipeline consists of a model to detect a decoy Vs. non-decoy and in addition we also perform a label reconstruction attack to reconstruct the raw labels of the client. The splinters are generated only using raw features. We see a spin-off empirical benefit that upon adding decoy splinters, not only do we prevent their detection; but we also make it extremely hard for the attacker to be able to reconstruct the raw labels corresponding to the raw data; via a second model.

# A    Upper bounds on KL-divergence between Gaussian mixtures

[16] defines the upper and lower bounds for KL-Divergence between GMMs to be:

$$D_{\text{lower}}(f||g) = \sum_a \omega_a^f \log \frac{\sum_\alpha \omega_\alpha^f e^{-D_{KL}(f_a||f_\alpha)}}{\sum_b \omega_b^g t_{ab}} - \sum_a \omega_a^f H(f_a) \tag{10}$$

$$D_{\text{upper}}(f||g) = \sum_a \omega_a^f \log \frac{\sum_\alpha \omega_\alpha^f z_{a\alpha}}{\sum_b \omega_b^g e^{-D_{KL}(f_a||g_b)}} + \sum_a \omega_a^f H(f_a) \tag{11}$$

where $H(f_a)$ is the entropy of $f_a$, and the normalization constants of the product of the individual Gaussians are given by:

$$\log t_{ab} = -\frac{d}{2} \log 2\pi - \frac{1}{2} \log |\Sigma_a^f + \Sigma_b^g| - \frac{1}{2}(\mu_b^g - \mu_a^f)^T (\Sigma_a^f + \Sigma_b^g)^{-1} (\mu_b^g - \mu_a^f) \tag{12}$$

$$\log z_{a\alpha} = -\frac{d}{2} \log 2\pi - \frac{1}{2} \log |\Sigma_a^f + \Sigma_\alpha^f| - \frac{1}{2}(\mu_\alpha^f - \mu_a^f)^T (\Sigma_a^f + \Sigma_\alpha^f)^{-1} (\mu_\alpha^f - \mu_a^f) \tag{13}$$

We will focus on optimizing the following average of the lower and upper bounds of the KL-Divergence between GMMs as it was shown to be a good estimate of the KL-Divergence between GMMs in [16].

$$D_{\text{avg}}(f||g) = \frac{1}{2}\left(D_{\text{lower}}(f||g) + D_{\text{upper}}(f||g)\right)$$

$$= \frac{1}{2}\left(\sum_a \omega_a^f \log \frac{\sum_\alpha \omega_\alpha^f e^{-D_{KL}(f_a||f_\alpha)}}{\sum_b \omega_b^g t_{ab}} - \sum_a \omega_a^f H(f_a)\right)$$

$$+ \frac{1}{2}\left(\sum_a \omega_a^f \log \frac{\sum_\alpha \omega_\alpha^f z_{a\alpha}}{\sum_b \omega_b^g e^{-D_{KL}(f_a||g_b)}} + \sum_a \omega_a^f H(f_a)\right)$$

$$= \frac{1}{2}\sum_a \omega_a^f \log \frac{\sum_\alpha \omega_\alpha^f e^{-D_{KL}(f_a||f_\alpha)}}{\sum_b \omega_b^g t_{ab}} + \frac{1}{2}\sum_a \omega_a^f \log \frac{\sum_\alpha \omega_\alpha^f z_{a\alpha}}{\sum_b \omega_b^g e^{-D_{KL}(f_a||g_b)}}$$

$$D_{\text{avg}}(f||g) = \frac{1}{2}\sum_a \omega_a^f \left[\log \sum_\alpha \omega_\alpha^f e^{-D_{KL}(f_a||f_\alpha)} + \log \sum_\alpha \omega_\alpha^f z_{a\alpha} - \log \sum_b \omega_b^g t_{ab} - \log \sum_b \omega_b^g e^{-D_{KL}(f_a||g_b)}\right]$$

If we assume that the data is mean-centered, the normalization constant $t_{ab}$ becomes

$$\log t_{ab} = -\frac{d}{2}\log 2\pi - \frac{1}{2}\log|\Sigma_a^f + \Sigma_b^g| = e^{\left(-\frac{d}{2}\log 2\pi - \frac{1}{2}\log|\Sigma_a^f + \Sigma_b^g|\right)} = (2\pi)^{-\frac{d}{2}}|\Sigma_a^f + \Sigma_b^g|^{-\frac{1}{2}}$$

Similarly, $z_{a\alpha} = (2\pi)^{-\frac{d}{2}}|\Sigma_a^f + \Sigma_\alpha^f|^{-\frac{1}{2}}$.

Plugging this into $(8)$, we get:

$$D_{\text{avg}}(f||g) = \frac{1}{2}\sum_a \omega_a^f \left[\begin{array}{l}\log \sum_\alpha \omega_\alpha^f e^{-D_{KL}(f_a||f_\alpha)} + \log \sum_\alpha \omega_\alpha^f (2\pi)^{-\frac{d}{2}}|\Sigma_a^f + \Sigma_\alpha^f|^{-\frac{1}{2}} \\ \qquad - \log \sum_b \omega_b^g (2\pi)^{-\frac{d}{2}}|\Sigma_a^f + \Sigma_b^g|^{-\frac{1}{2}} - \log \sum_b \omega_b^g e^{-D_{KL}(f_a||g_b)}\end{array}\right]$$

$$= \frac{1}{2}\sum_a \omega_a^f \left[\begin{array}{l}\log \sum_\alpha \omega_\alpha^f e^{-D_{KL}(f_a||f_\alpha)} + \frac{d\log 2\pi}{2} + \log \sum_\alpha \frac{\omega_\alpha^f}{\sqrt{|\Sigma_a^f + \Sigma_\alpha^f|}} \\ \qquad - \frac{d\log 2\pi}{2} - \log \sum_b \frac{\omega_b^g}{\sqrt{|\Sigma_a^f + \Sigma_b^g|}} - \log \sum_b \omega_b^g e^{-D_{KL}(f_a||g_b)}\end{array}\right]$$

$$D_{\text{avg}}(f||g) = \frac{1}{2}\sum_a \omega_a^f \left[\log \sum_\alpha \omega_\alpha^f e^{-D_{KL}(f_a||f_\alpha)} + \log \sum_\alpha \frac{\omega_\alpha^f}{\sqrt{|\Sigma_a^f + \Sigma_\alpha^f|}} - \log \sum_b \omega_b^g e^{-D_{KL}(f_a||g_b)} - \log \sum_b \frac{\omega_b^g}{\sqrt{|\Sigma_a^f + \Sigma_b^g|}}\right]$$

$$(14)$$

## B Scroll down for modified EM updates for covariance and mean that we optimize via Powell minimization

$$\mu_b^{g\,(t+1)} =$$

$$\min_{\mu} \left\{ \begin{array}{l} -\omega_b^f \log\left[\sum_{b'\neq b}\omega_{b'}^{g\,(t)}e^{\mathrm{DCov}\left(\Sigma_{b'}^f,\frac{1}{N-1}\left(Z_{b'}^{(t)}-\mu_{b'}^{g\,(t)}\right)^T\left(Z_{b'}^{(t)}-\mu_{b'}^{g\,(t)}\right)\right)} + \omega_b^{g(t)}e^{\mathrm{DCov}\left(\Sigma_b^f,\frac{1}{N-1}\left(Z_b^{(t)}-\mu\right)^T\left(Z_b^{(t)}-\mu\right)\right)}\right] \\[4em]

-\omega_b^f \log\left[\sum_{b'\neq b}\dfrac{\omega_{b'}^{g\,(t)}e^{-\frac{1}{2}\left(\mu_{b'}^g-\mu_{b'}^f\right)^T\left(\Sigma_{b'}^f+\frac{1}{N-1}\left(Z_{b'}^{(t)}-\mu_{b'}^{g\,(t)}\right)^T\left(Z_{b'}^{(t)}-\mu_{b'}^{g\,(t)}\right)\right)^{-1}\left(\mu_{b'}^g-\mu_{b'}^f\right)}}{\sqrt{\left|\Sigma_{b'}^f+\frac{1}{N-1}\left(Z_{b'}^{(t)}-\mu_{b'}^{g\,(t)}\right)^T\left(Z_{b'}^{(t)}-\mu_{b'}^{g\,(t)}\right)\right|}} \right. \\[3em]

\left. +\dfrac{\omega_b^{g(t)}e^{-\frac{1}{2}\left(\mu-\mu_b^f\right)^T\left(\Sigma_b^f+\frac{1}{N-1}\left(Z_b^{(t)}-\mu\right)^T\left(Z_b^{(t)}-\mu\right)\right)^{-1}\left(\mu-\mu_b^f\right)}}{\sqrt{\left|\Sigma_b^f+\frac{1}{N-1}\left(Z_b^{(t)}-\mu\right)^T\left(Z_b^{(t)}-\mu\right)\right|}}\right] \\[3em]

+\omega_b^f(C_1+C_2) \\[2em]

+\dfrac{1}{2}\sum_{b'\neq b}\left[ n_{b'}^{(t+1)}\log\left|\left(\frac{1}{N-1}\left(Z_{b'}^{(t)}-\mu_{b'}^{g\,(t)}\right)^T\left(Z_{b'}^{(t)}-\mu_{b'}^{g\,(t)}\right)\right)^{-1}\right| \right. \\[2em]

\left. +\sum_{i=1}^N \gamma_{ib'}^{(t+1)}\mathrm{Tr}\left(\left(\frac{\left(Z_{b'}^{(t)}-\mu_{b'}^{g\,(t)}\right)^T\left(Z_{b'}^{(t)}-\mu_{b'}^{g\,(t)}\right)}{N-1}\right)^{-1}\left(x_i-\mu_{b'}^{g\,(t)}\right)\left(x_i-\mu_{b'}^{g\,(t)}\right)^T\right)\right] \\[2em]

+\dfrac{1}{2}\left[ n_b^{(t+1)}\log\left|\left(\frac{1}{N-1}\left(Z_b^{(t)}-\mu\right)^T\left(Z_b^{(t)}-\mu\right)\right)^{-1}\right| \right. \\[2em]

\left. +\sum_{i=1}^N \gamma_{ib}^{(t+1)}\mathrm{Tr}\left(\left(\frac{\left(Z_b^{(t)}-\mu\right)^T\left(Z_b^{(t)}-\mu\right)}{N-1}\right)^{-1}\left(x_i-\mu\right)\left(x_i-\mu\right)^T\right)\right] \end{array}\right\}$$

We use Powell minimization method to optimize for $\mathbf{Z_b}$ as

$$Z_b^{(t+1)} =$$

$$\min_Z \left\{ \begin{array}{l} \left[ -\omega_b^f \log \left[ \sum_{b' \neq b} \omega_{b'}^{g\,(t)} e^{\text{DCov}\left( \Sigma_{b'}^f, \frac{1}{N-1} \left( Z_{b'}^{(t)} - \mu_{b'}^{g\,(t)} \right)^T \left( Z_{b'}^{(t)} - \mu_{b'}^{g\,(t)} \right) \right)} + \omega_b^{g(t)} e^{\text{DCov}\left( \Sigma_b^f, \frac{1}{N-1} \left( Z - \mu_b^{g\,(t)} \right)^T \left( Z - \mu_b^{g\,(t)} \right) \right)} \right] \right. \\[4mm] \left. -\omega_b^f \log \left[ \sum_{b' \neq b} \frac{\omega_{b'}^{g\,(t)} e^{-\frac{1}{2}\left( \mu_{b'}^g - \mu_{b'}^f \right)^T \left( \Sigma_{b'}^f + \frac{1}{N-1}\left( Z_{b'}^{(t)} - \mu_{b'}^{g\,(t)} \right)^T \left( Z_{b'}^{(t)} - \mu_{b'}^{g\,(t)} \right) \right)^{-1} \left( \mu_{b'}^g - \mu_{b'}^f \right)}}{\sqrt{\left| \Sigma_{b'}^f + \frac{1}{N-1} \left( Z_{b'}^{(t)} - \mu_{b'}^{g\,(t)} \right)^T \left( Z_{b'}^{(t)} - \mu_{b'}^{g\,(t)} \right) \right|}} \right.\right. \\[4mm] \left.\left. + \frac{\omega_b^{g(t)} e^{-\frac{1}{2}\left( \mu_b^{g\,(t)} - \mu_b^f \right)^T \left( \Sigma_b^f + \frac{1}{N-1}\left( Z - \mu_b^{g\,(t)} \right)^T \left( Z - \mu_b^{g\,(t)} \right) \right)^{-1} \left( \mu_b^{g\,(t)} - \mu_b^f \right)}}{\sqrt{\left| \Sigma_b^f + \frac{1}{N-1} \left( Z - \mu_b^{g\,(t)} \right)^T \left( Z - \mu_b^{g\,(t)} \right) \right|}} \right] \right. \\[4mm] \left. + \omega_b^f (C_1 + C_2) \right. \\[4mm] \left. + \frac{1}{2} \sum_{b' \neq b} \left[ n_{b'}^{(t+1)} \log \left| \left( \frac{1}{N-1} \left( Z_{b'}^{(t)} - \mu_{b'}^{g\,(t)} \right)^T \left( Z_{b'}^{(t)} - \mu_{b'}^{g\,(t)} \right) \right)^{-1} \right| \right.\right. \\[4mm] \left.\left. + \sum_{i=1}^N \gamma_{ib'}^{(t+1)} \text{Tr} \left( \left( \frac{\left( Z_{b'}^{(t)} - \mu_{b'}^{g\,(t)} \right)^T \left( Z_{b'}^{(t)} - \mu_{b'}^{g\,(t)} \right)}{N-1} \right)^{-1} \left( x_i - \mu_{b'}^{g\,(t)} \right) \left( x_i - \mu_{b'}^{g\,(t)} \right)^T \right) \right] \right. \\[4mm] \left. + \frac{1}{2} \left[ n_b^{(t+1)} \log \left| \left( \frac{1}{N-1} \left( Z - \mu_b^{g(t)} \right)^T \left( Z - \mu_b^{g(t)} \right) \right)^{-1} \right| \right.\right. \\[4mm] \left.\left. + \sum_{i=1}^N \gamma_{ib}^{(t+1)} \text{Tr} \left( \left( \frac{\left( Z - \mu_b^{g(t)} \right)^T \left( Z - \mu_b^{g(t)} \right)}{N-1} \right)^{-1} \left( x_i - \mu_b^{g(t)} \right) \left( x_i - \mu_b^{g(t)} \right)^T \right) \right] \right. \end{array} \right\}$$

## C   Proof of Theorem 4.1

*Proof.* This condition simplifies to requiring

$$\sqrt{|\Sigma_a^f + \Sigma_b^g|} \leq \omega_b^g, \forall b$$

By the arithmetic-geometric-mean (A.G.M) inequality we have,

$$\prod_{k=1}^n \lambda_k \leq \frac{1}{n^n} \left( \sum_{k=1}^n \lambda_k \right)^n$$

Therefore $\sum_b |\Sigma_a^f + \Sigma_b^g| \leq \frac{\sum_b [\text{Tr}(\Sigma_a^f + \Sigma_b^g)]^n}{n^n}$ This implies that if,

$$\sum_b \text{Tr}(\Sigma_a^f + \Sigma_b^g) \leq n \sqrt[n]{\omega_b^g}, \forall b$$

then the condition for convexity $\sum_b \sqrt{|\Sigma_a^f + \Sigma_b^g|} \leq n \sqrt[n]{\omega_b^g}, \forall b$ will be satisfied. $\square$

## D   Proof of Theorem 4.2

*Proof.* We now show that the LogSumExp function $\log \sum_b \omega_b^g e^{\text{DCov}(\Sigma_f^a, \Sigma_b^g)}$ is convex as well. In fact, $LogSumExp(f(z))$ happens to be convex for any convex function $f(z)$ as shown below.

$$\frac{\partial^2}{\partial z^2} log \sum e^{f_i(z)} = \frac{\partial}{\partial z} \left[ \frac{\sum(e_i^f(z) \frac{\partial}{\partial z} f_i(z))}{\sum e^{f_i(z)}} \right] \tag{15}$$

which is equal to

$$\frac{\sum e_i^f \frac{\partial^2}{\partial z^2} f_i(z)}{\sum e^{f_i(z)}} + \frac{\sum e^{f^i(z)} [\frac{\partial}{\partial z} f_i(z)]^2}{\sum e^{f_i(z)}} - \frac{(\sum e^{f_i(z)} \frac{\partial}{\partial z} f_i(z))^2}{(\sum e^{f_i(z)})^2} \tag{16}$$

The first term is positive. The difference of the next two terms is positive due to Jensen's inequality as

$$\sum \left[ a_i \left( \frac{\partial}{\partial z} f_i(z) \right)^2 \right] \geq \left[ \sum a_i \frac{\partial}{\partial z} f_i(z) \right]^2 \tag{17}$$

This proves convexity of $\log \sum_b \omega_b^g e^{\text{DCov}(\Sigma_f^a, \Sigma_b^g)}$.

$\square$

# E   Upper and lower bounds on distance correlation

# F   Lower bound

*Proof.*
$$\det(\mathbf{Z^T X}) - \det(\mathbf{Z^T Z}) - \det(\mathbf{X^T Z}) + \det(\mathbf{X^T X})$$

can be bounded using Hadamard's inequality as

$$\det(\mathbf{Z^T X}) - \det(\mathbf{Z^T Z}) + \det(\mathbf{X^T X}) - \det(\mathbf{X^T Z})$$

$$\leq \mathbf{Z^T X} - \mathbf{Z^T Z}_2 \frac{\mathbf{Z^T X}_2^n - \mathbf{Z^T Z}_2^n}{\mathbf{Z^T X}_2 - \mathbf{Z^T Z}_2}$$

$$+ \mathbf{X^T Z} - \mathbf{X^T X}_2 \frac{\mathbf{X^T Z}_2^n - \mathbf{X^T X}_2^n}{\mathbf{X^T Z}_2 - \mathbf{X^T X}_2}$$

The fractional terms $\frac{\mathbf{Z^T X}_2^n - \mathbf{Z^T Z}_2^n}{\mathbf{Z^T X}_2 - \mathbf{Z^T Z}_2}, \frac{\mathbf{X^T Z}_2^n - \mathbf{X^T X}_2^n}{\mathbf{X^T Z}_2 - \mathbf{X^T X}_2}$ can be written as a sum of geometric-series, with factors of change of $\frac{\mathbf{Z^T X}}{\mathbf{Z^T Z}}, \frac{\mathbf{X^T Z}}{\mathbf{X^T X}}$ respectively because

$$\frac{\mathbf{Z^T X}_2^n - \mathbf{Z^T Z}_2^n}{\mathbf{Z^T X}_2 - \mathbf{Z^T Z}_2} = \frac{1 - (\frac{\mathbf{Z^T X}_2}{\mathbf{Z^T Z}_2})^n}{1 - \frac{\mathbf{Z^T X}_2}{\mathbf{Z^T Z}_2}}$$

$$= \sum_{p=0}^{n-1} \mathbf{Z^T X}_2^p \mathbf{Z^T Z}_2^{p-1}$$

Therefore these fractional terms can be minimized by minimizing $\mathbf{Z^T X}_2$ and $\mathbf{Z^T Z}_2$ as the sums of products of decreasing functions of norms are also decreasing. By Cauchy-Schwarz inequality $\mathbf{Z^T}(\mathbf{X} - \mathbf{Z}) \leq \mathbf{ZX} - \mathbf{Z}$.

Therefore minimizing $\mathbf{Z}$ and $\mathbf{X} - \mathbf{Z}$ to minimize terms $\mathbf{Z^T X} - \mathbf{Z^T Z}, \mathbf{X^T Z} - \mathbf{X^T X}$ in addition to minimizing $\mathbf{Z^T Z}$, $\mathbf{Z^T X}_2 = Tr(\mathbf{Z^T X X^T Z}) = DCOV(\mathbf{X}, \mathbf{Z})$ minimizes terms $\frac{\mathbf{Z^T X}_2^n - \mathbf{Z^T X}_2^n}{\mathbf{Z^T X}_2 - \mathbf{Z^T Z}_2}$, $\frac{\mathbf{X^T Z}_2^n - \mathbf{X^T X}_2^n}{\mathbf{X^T Z}_2 - \mathbf{X^T X}_2}$ which gives us the desired result. $\square$

**Our upper bound:**

*Proof.* Based on the definition of Lipschitz continuity we have the following bound where $L$ is the Lipschitz constant of the map that learns $\mathbf{Z}$ from $\mathbf{X}$,

$$f(\mathbf{X_i}) - f(\mathbf{X_j})^2 = \mathbf{Z_i} - \mathbf{Z_j}^2 \leq L\mathbf{X_i} - \mathbf{X_j}^2 \tag{18}$$

Multiplying by $\langle \mathbf{X_i}, \mathbf{X_j} \rangle$ on both sides and summing over all points we have

$$\sum_{ij} \mathbf{Z_i} - \mathbf{Z_j}^2 \langle \mathbf{X_i}, \mathbf{X_j} \rangle \leq L \sum_{ij} \mathbf{X_i} - \mathbf{X_j}^2 \langle \mathbf{X_i}, \mathbf{X_j} \rangle$$

Now dividing on both sides by

$\sqrt{\sum_{ij} \mathbf{Z_i} - \mathbf{Z_j}^2 \langle \mathbf{Z_i}, \mathbf{Z_j} \rangle} \sqrt{\sum_{ij} \mathbf{X_i} - \mathbf{X_j}^2 \langle \mathbf{X_i}, \mathbf{X_j} \rangle}$ we get

$$DCOR(\mathbf{X}, \mathbf{Z}) \leq \frac{L\sqrt{\sum_{ij} \mathbf{X_i} - \mathbf{X_j}^2 \langle \mathbf{X_i}, \mathbf{X_j} \rangle}}{\sqrt{\sum_{ij} \mathbf{Z_i} - \mathbf{Z_j}^2 \langle \mathbf{Z_i Z_j} \rangle}} \tag{19}$$

But $\frac{\sqrt{\sum_{ij} \mathbf{X_i} - \mathbf{X_j}^2 \langle \mathbf{X_i}, \mathbf{X_j} \rangle}}{\sqrt{\sum_{ij} \mathbf{Z_i} - \mathbf{Z_j}^2 \langle \mathbf{Z_i Z_j} \rangle}}$ is the ratio of distance standard deviations which is the square root of distance variance which is inturn distance covariance between a variable and itself. It has been shown in [17] that the distance standard deviation can be upper bounded by the trace of the covariance matrix. Therefore we have

$$DCOR(\mathbf{X}, \mathbf{Z}) \leq \frac{L.Tr(\Sigma_{\mathbf{X}})}{\sqrt{\sum_{ij} \mathbf{Z_i} - \mathbf{Z_j}^2 \langle \mathbf{Z_i Z_j} \rangle}} \tag{20}$$

and similarly

$$DCOV(\mathbf{X}, \mathbf{Z}) \leq L.[Tr(\Sigma_{\mathbf{X}})]^2 \tag{21}$$

Therefore combining our sample SIV inequality with a concentration Hoeffding bound on the quality of estimating population distance covariance from sample distance covariance in [18] we get with high-probability $1 - \delta$ an updated bound of

$$DCOV(p_{xy}, \mathcal{F}, \mathcal{G}) \pm \epsilon \leq \sqrt{\frac{log(6/\delta)}{0.24n}} + \frac{C}{n} + L.[Tr(\Sigma_{\mathbf{X}})]^2 \tag{22}$$

□

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
