# OpenReview forum: "Sniper GMMs: Structured Gaussian mixtures poison ML on large n small p data with high efficacy"
_NeurIPS.cc/2020/Workshop/DL-IG — Submitted to NeurIPSW 2020: DL-IG_

### Official Review · AnonReviewer1 · 2020-10-22
**Review of "Sniper GMMs: Structured Gaussian mixtures poison ML on large n small p data with high efficacy "**

**Rating:** 2
**Confidence:** 4

**Review:**

In Theorem 2.1, the authors appear to have used KL-divergence for the term h(X,Z)- h(Z) = h(X|Z). Although h(X|Z) is called the relative entropy in information theory (KL-divergence is also called the relative entropy in probability), these two terms are different. To emphasize, h(X|Z) is not equal to KL(p_X || p_Z) = \int p_X \log p_Z/p_X. KL divergence does not depend on the joint distribution of (X, Z), only on the marginals. Relative entropy h(X|Z) does depend on the marginals. Please fix this as it is likely to confuse all readers.

The KL-divergence bound at the bottom of page two is not proved in Appendix B, as claimed in the paper. After some searching, it appears to be in Appendix A.  Appendix E only has a title, and appendix G is entirely missing.

The paper is simply not ready to submit anywhere. Given the confusion regarding KL divergence in the main theorem, I would advise the authors to carefully check if their approach is still valid.

---

### Official Review · AnonReviewer2 · 2020-10-27
**Wrong definition of KL and its usage, lack of references and typos**

**Rating:** 3
**Confidence:** 5

**Review:**

The topic of the paper is interesting. However, it seems that the paper has been wrapped up quickly and present some errors.
For example,
In Theorem 2.1: missing notation for distance correlation argmin_Z(X,Z).

It is important to distinguish between distance between random variables (eg, Mutual information) and distances between distributions (eg, KL).
In the proof of 2.1, the Authors use a wrong definition for KL.
The paper is not yet ready for communication as it lacks references and need to correct the definition of KL and its usage.

- For structured GMMs, I recommend
Gaussian parsimonious clustering models
https://www.sciencedirect.com/science/article/abs/pii/0031320394001256

- State also that GMMs are universal smooth density approximators

- For outliers contamination, you may be interested by
Robust parameter estimation with a small bias against heavy contamination
https://www.sciencedirect.com/science/article/pii/S0047259X08000456

- For bounds on KL between GMMs (or any other joint convex statistical distance), we can relax by a Linear Program as described in
On The Chain Rule Optimal Transport Distance
https://arxiv.org/abs/1812.08113

---

### Decision · Program_Chairs · 2020-11-07

**Decision:**

Reject

**Comment:**

Dear authors, please see the reviews below. The reviewers discuss how the definition of the KL divergence used in the paper is really the mutual information. The manuscript, as written, is not ready for dissemination and you are strongly encouraged to use this feedback to improve it.